# MRI Tumor Regression Grade Combined with T2-Weighted Volumetry May Predict Histopathological Response in Locally Advanced Rectal Cancer following Neoadjuvant Chemoradiotherapy—A New Scoring System Proposal

**DOI:** 10.3390/diagnostics13203226

**Published:** 2023-10-17

**Authors:** Aleksandra Jankovic, Jelena Djokic Kovac, Marko Dakovic, Milica Mitrovic, Dusan Saponjski, Ognjen Milicevic, Aleksandra Djuric-Stefanovic, Goran Barisic

**Affiliations:** 1Department for Digestive Radiology, Center for Radiology, University Clinical Center of Serbia, Pasterova No. 2, 11000 Belgrade, Serbia; jelenadjokickovac@gmail.com (J.D.K.); dr_milica@yahoo.com (M.M.); saponjski.d@gmail.com (D.S.); aleksandra.djuricstefanovic@gmail.com (A.D.-S.); 2Faculty of Medicine, University of Belgrade, Dr. Subotica No. 8, 11000 Belgrade, Serbia; ognjen011@gmail.com (O.M.); barisic_goran@yahoo.com (G.B.); 3Faculty of Physical Chemistry, University of Belgrade, 11000 Belgrade, Serbia; marko@ffh.bg.ac.rs; 4Clinic for Digestive Surgery—First Surgical Clinic, University Clinical Center of Serbia, Koste Todorovica No. 6, 11000 Belgrade, Serbia

**Keywords:** magnetic resonance imaging, responders, non-responders, rectal cancer, magnetic resonance volumetry

## Abstract

Modern studies focus on the discovery of innovative methods to improve the value of post-treatment magnetic resonance imaging (MRI) in the prediction of pathological responses to preoperative neoadjuvant chemoradiotherapy (nCRT) in locally advanced rectal cancer (LARC)**.** The aim of this study was to assess the potential benefits of combining magnetic resonance tumor regression grade (mrTRG) with T2-weighted volumetry in the prediction of pathological responses to nCRT in LARC. This was a cohort study conducted on patients with histopathologically confirmed LARC in a period from 2020 to 2022. After histopathological verification, all patients underwent initial MRI studies, while the follow-up MRI was performed after nCRT. Tumor characteristics, MRI estimated tumor regression grade (mrTRG) and tumor volumetry were evaluated both initially and at follow-up. All patients were classified into responders and non-responders according to pathological tumor regression grade (pTRG) and mrTRG. A total of 71 patients, mostly male (66.2%) were included in the study. The median tumor volume reduction rate was significantly higher in nCRT-responders compared to non-responders (79.9% vs. 63.3%) (*p* = 0.003). Based on ROC analysis, optimal cut-off value for tumor volume reduction rate was determined with an area under the curve (AUC) value of 0.724 (*p* = 0.003). Using the tumor volume reduction rate ≥75% with the addition of response to nCRT according to mrTRG, a new scoring system for prediction of pTRG to preoperative nCRT in LARC was developed. Diagnostic performance of prediction score was tested and the sensitivity, PPV, specificity, and NPV were 81.8%, 56.3%, 71.4%, and 89.7%, respectively. The combination of mrTRG and T2-weighted volumetry increases the MRI-based prediction of pTRG to preoperative nCRT in LARC. The proposed scoring system could aid in distinguishing responders to nCRT, as these patients could benefit from organ-preserving treatment and a “watch and wait” strategy.

## 1. Introduction

Colorectal cancer is the second most prevalent cancer in women and the third most common in men, with approximately 44% of total cases localized in the rectum [1]. Significantly increased prevalence of rectal cancer can be attributed to the modern lifestyle, characterized by poor nutrition and lack of physical activity. However, improved screening coverage, together with considerable advances in diagnosis and current treatment approaches, have led to earlier cancer detection and, in general, prolonged survival [2]. Locally advanced rectal cancer (LARC) is defined as a tumor of the clinical stage T3/T4 or a tumor of any stage with positive lymphonodal status [3]. In modern conditions, the standard procedure for treating patients with LARC is the application of neoadjuvant chemoradiotherapy (nCRT) as an initial therapeutic option [4,5,6]. This treatment strategy reduces risk factors for metastatic disease such as tumor mass, lymphadenopathy, and extramural vascular invasion.

MRI is the imaging procedure which provides excellent insight into the anatomy of the rectal wall, mesorectal fat and other pelvic structures. Therefore, MRI is the preferred diagnostic modality for the initial examination of patients with histopathologically proven rectal cancer. By determining the local stage of the primary tumor, as well as the involvement of loco-regional lymph nodes, MRI examination plays a key role in selecting patients with rectal cancer who could benefit from nCRT. A post-treatment MRI examination is usually performed six to eight weeks after nCRT. After treatment, the role of MRI examination is to determine the degree of tumor regression by analogy with the histopathological degree of tumor regression (tumor regression grade—TRG), which represents the gold standard [7]. The response to the therapy is classified on a scale from TRG-1 (visible fibrosis only, probably complete response) to TRG-5 (no fibrosis, mostly residual disease) [7,8]. It is important to note that, unlike the extraordinary accuracy of the initial pre-treatment MRI examination, the reliability of interpretation of the post-treatment examination decreases dramatically, as it is often hard or even impossible to make a difference between signal intensities of the therapy-induced fibrosis and viable tumor cells [9]. That is the reason why other MRI parameters besides those included in the standard protocol are needed in everyday clinical practice for assessment of nCRT response in patients with rectal cancer.

One of the advanced MRI techniques is volumetric analysis, whose value is not yet thoroughly analyzed in patients with LARC. In this setting, Martens et al. highlighted the importance of this analysis, and showed that the difference in tumor volume between initial and post-treatment MRI examinations has a great predictive potential, with diagnostic accuracy ranging from 71% to 73% [10].

To the best of our knowledge, there were no previous studies that evaluated the utility of combination of MRI-based T2-weighted volumetry and mrTRG for prediction of pathological response in LARC. It could be of crucial importance if responders (patients with complete and near complete responses) could be accurately distinguished from non-responders, as the responders could benefit from organ-preserving treatment and a “watch and wait” strategy [11,12]. Therefore, the aim of this study was to investigate the potential benefits of combining MRI-based T2-weighted volumetry and mrTRG in the prediction of pathological response in patients with LARC after receiving nCRT and to propose the new MRI-based scoring system.

## 2. Materials and Methods

This was a cohort study which included seventy-one patients with histopathologically confirmed rectal cancer in a period from January 2020 to December 2022. The study was approved by the relevant institutional review board, and written informed consent was obtained from all patients.

The inclusion criteria for the patients were as follows: (1) histopathological confirmation of rectal cancer; (2) patients treated with neoadjuvant chemoradiotherapy; (3) both initial and control MRI examination performed; and (4) patients who underwent surgery after neoadjuvant chemoradiotherapy. Exclusion criteria were: (1) patients younger than 18 years; (2) patients who had contraindications for MR examination (e.g., metal implants); (3) absence of operative treatment; (4) technically inadequate quality of any of the mentioned MR examinations due to the presence of artifacts (e.g., motion artifacts) as interpretation of those MRI sequences is not accurate; and (5) evidence of metastases at the time of diagnosis.

Within two weeks after histopathological verification, all patients underwent an initial MRI examination. Post-treatment MRI examination was performed six to eight weeks after chemoradiotherapy, in accordance with widely accepted protocols. It is generally advised to perform surgery after the stated time frame. This could be explained by the fact that many previous studies have shown that this period of time is necessary for the radiation therapy to exert an effect. Moreover, some side effects of radiation therapy, such as rectal wall edema, could impair the appropriate interpretation of nCRT response [13,14].

### 2.1. MRI Technique

In preoperative diagnostics, both initial and post-treatment MRI examinations at 1.5-T (Signa HDxt, GE Healthcare, Waukesha, WI, USA) were performed using the same protocol, which included: standard T1-weighted and T2-weighted sequences, high-resolution T2-weighted sequences in the coronary and axial planes angled along the long axis of the tumor; and diffusion-weighted imaging (DWI) with different b values (*b* = 0 s/mm^2^, *b* = 800 s/mm^2^) from which the apparent diffusion coefficient (ADC) maps were automatically calculated.

The MRI examinations were reviewed in consensus by two radiologists with 8 and 13 years of expertise in rectal cancer MRI. The following parameters were analyzed: tumor localization, craniocaudal tumor diameter, presence of extramural tumor propagation, lympho-nodal status, assessment of tumor infiltration of the mesorectal fascia (MRF), assessment of extramural vascular invasion according to Smith, local tumor stage before and after nCRT, and MRI estimated tumor regression grade (mrTRG). We used a scoring system established by the Magnetic Resonance Imaging and Rectal Cancer European Equivalence Study (MERCURY) study group for mrTRG. The construction of mrTRG is based on pathologic TRG (pTRG). The classification of mrTRG relies on the relative prevalence of fibrous or tumor signal intensity within the tumor entirety. In the context of radiology, Grade 1 signifies a comprehensive radiologic response characterized by the presence of a linear or crescentic scar, as observed on MRI. Grade 2 denotes a favorable response, wherein MRI findings reveal dense fibrosis without any discernible residual tumor, thereby suggesting either minimal residual disease or the absence of a tumor. Grade 3 corresponds to a moderate response, wherein more than 50% of the areas exhibit fibrosis or mucin, accompanied by visible intermediate tumor signal on MRI. Grade 4 indicates a minimal response to treatment, with MRI findings indicating the presence of a few areas with fibrosis or mucin, predominantly displaying tumor-derived MRI signals. Lastly, Grade 5 signifies a lack of response to therapy, characterized by a tumor similar to the baseline or significant regrowth of the tumor [15].

The degree of tumor response to applied therapy was evaluated on the basis of standard and targeted T2-weighted sequences, DWI parameters, and ADC values. Firstly, the tumor was identified and evaluated using abovementioned sequences. Secondly, volumetry was measured on both the initial and post-treatment MRI examination using the Medical Image Processing, Analysis, and Visualization (MIPAV) application, in which a volume of interest (VOI) was created by manually drawing a region of interest (ROI) on each tumor slice, and the tumor volume was determined. Each patient’s images were evaluated separately by a radiologist who was blinded to the patient’s histopathological report.

Histopathological findings were considered the gold standard for assessing the diagnostic accuracy of certain MRI examination methods. It included defining tumor regression grade according to Mandarat (pTRG; TRG 1—complete response, TRG 2—near complete response with dense fibrosis, TRG 3—moderate response with more than 50% of fibrosis, TRG 4—slight response with subtle areas of fibrosis but predomination of tumor tissue, TRG 5—no response) [7]. Finally, radiologists classified all patients into two groups: responders (pTRG 1,2) and non-responders (pTRG 3–5).

### 2.2. Statistical Analysis

The descriptive statistics, including means, medians, standard deviations and percentiles for numerical variables, and numbers and percentages for categorical variables were used to characterize the study sample. The Mann–Whitney U test was used for numerical data without normal distribution to evaluate the differences between responders and non-responders after nCRT. C statistic, representing the area under the receiver operating characteristic curve, was used for overall assessment of the predictive model. Model discrimination performance was tested by means of sensitivity, specificity, positive, and negative predictive values. Sensitivity was defined as the % of patients who have score ≥1 among pTRG responders. Specificity was defined as the % of patients who have score <1 among pTRG non-responders. The positive predictive value was defined as the % of pTRG responders among patients who have score ≥1. A negative predictive value was defined as the % of pTRG non-responders among patients who have score <1. In all analyses, level of statistical significance was set at *p* ≤ 0.05. For the statistical analysis, the SPSS version 25 statistical software (Chicago, IL, USA) was used.

## 3. Results

### 3.1. Patients

A total of 71 patients with histopathologically confirmed rectal cancer were included in the study. The average age of study participants was 61.5 ± 11.4 years and more than half were male (66.2%). Detailed demographic, clinical and MRI imaging characteristics of study population are presented in Table 1.

### 3.2. Diagnostic Performance of mrTRG

According to mrTRG there were 18 (25.4%) patients with mrTRG1-2 and 53 patients with mrTRG3-4 (74.6%), whereas according to pTRG there were 22 patients (31.0%) with pTRG 1–2 and 49 patients with pTRG 3–5 (69.0%). Diagnostic performance of mrTRG in terms of sensitivity, specificity, PPV and NPV were 50%, 85.7%, 61.7% and 79.2%, respectively. MRF invasion after nCRT was present in 14 patients (19.7%). In non-responders, MRF invasion was more frequently present (26.5%) in comparison to responders (4.5%) (Table 2).

### 3.3. Tumor Volume for Prediction of nCRT Response

The median tumor volume before nCRT was 12,998.0 (25th–75th percentile, 9920.4–24,552.3), while median volume after nCRT was 4670.3 (25th–75th percentile, 2249.0–9075.2) with median percent of tumor volume regression of 66.4% (25th–75th percentile, 50.1–81.0). In patients who had an adequate response to nCRT (responders) median tumor volume regression was 79.9% in comparison to the non-responder group of patients, where the median tumor volume regression was 63.3% (*p* = 0.003) (Table 3).

ROC curve analysis (Figure 1) was used to define optimal cut-off value of tumor volume regression for predicting pTRG response. Area under the curve (AUC) for tumor volume regression was 0.724 (*p* = 0.003). Diagnostic performance of different cut offs of tumor volume regression for predicting pTRG response in terms of sensitivity, specificity, PPV and NPV are presented in Table 4.

### 3.4. Neoadjuvant CRT Prediction Score

A new nCRT response prediction score was developed using combination of mrTRG and tumor volume regression grade (Table 5). The score was arrived at by assigning 0 points for tumor volume regression <75% and patients with mrTRG 3–5; 1 point was assigned if tumor volume regression was ≥75% and patients with mrTRG 3–5, or if tumor volume regression was <75% and patients with mrTRG 1,2; and 2 points were assigned if tumor volume regression was ≥75% and patients with mrTRG 1,2.

Based on the newly developed nCRT response prediction score, the study population was divided into the following categories: low response (score 0), medium response (score 1), and high response (score 2) to nCRT (Figure 2, Figure 3 and Figure 4). Sum of mrTRG and T2-volumetry improved AUC to 0.801. The distribution of the patients according to these groups is presented in Figure 5.

Diagnostic performance of newly developed prediction score is presented in Table 6.

## 4. Discussion

Our study aimed to introduce a novel scoring model to predict the response after neoadjuvant chemoradiotherapy in patients with LARC. The scoring system consisted of the percentage of tumor volume regression and the response to nCRT according to mrTRG. The results of our study showed a high chance of response prediction in patients with a tumor volume regression ≥75% and mrTRG grades 1 and 2. The diagnostic performance of the novel scoring system showed a sensitivity of 81.8% and specificity of 71.4%.

Several recent studies evaluated the value of tumor volume assessment for prediction of pathologic response following nCRT [16,17,18,19,20,21]. In this regard, Lutsyk et al. showed that the gross tumor volume <39.5 cm^2^ was the only predictive factor for achieving pathological complete response (CR) with area under ROC curve 0.715 for rectal adenocarcinoma stage II and 0.62 for stage III [16]. Moreover, initial tumor size greater than 3 cm and tumor volume have been associated with more aggressive tumor behavior and lower sensitivity to nCRT [17]. In a similar study, it was shown that tumors with the volume above the threshold of 37.3 cm^3^ had a probability of 78% and greater of not achieving pCR [18]. The significance of tumor volume measurement in LARC was also pointed out in the study by Neri et al. who showed that tumor volume reduction rate after nCRT, measured by MR volumetry, correlated well with pTRG [19]. Furthermore, Xiao et al. concluded that there was a significant correlation between tumor volume reduction rate measured by MRI and pTRG, along with overall downstage in LARC patients treated with preoperative chemotherapy alone [20]. Similar to the abovementioned studies, our results have shown high diagnostic accuracy of tumor volume reduction rate in the prediction of pathological response with sensitivity and specificity being 68.2% and 81.6%, respectively, for tumor volume regression grade more than 75%. In agreement with our findings, Nougaret et al. identified 70% as the optimal cut-off value of volume reduction for predicting excellent histologic response based on sensitivity of 86% and specificity of 100% [21]. Therefore, we could conclude that inclusion of T2-weighted volumetry and measurement of tumor volume reduction rate increases the potential significance of MRI in predicting the pathological response after nCRT.

Numerous previous studies evaluated the value of mrTRG in the prediction of pathological response after nCRT [10,22,23,24,25,26]. However, there is high variability in published results, raising concerns about the accuracy of MRI in restaging patients after nCRT. Therefore, the usefulness of mrTRG as a biomarker of nCRT response remains controversial. A recent study by Sclafani et al. conducted on 191 patients with rectal cancer, which compared the agreement between mrTRG and pTRG, found a fair accordance between the two in a standard five-tier regression classification with a sensitivity and specificity of mrTRG 1–2 in prediction of complete pathological response being 74.4% and 62.8%, respectively. However, ultimately, they showed a low agreement between mrTRG and pTRG and concluded that mrTRG cannot be used as a surrogate of pTRG [22]. Another similar study showed limited reproducibility of mrTRG among radiologists and a low agreement of mrTRG with pTRG [10]. In line with previous studies, our results have shown low sensitivity and specificity of mrTRG in predicting pTRG, being 50% and 85.7%, respectively. According to previous literature, one of the reasons for the low agreement between mrTRG and pTRG is inability of MRI to accurately distinguish residual tumor from fibrosis and desmoplastic reaction [23,24]. Additionally, the lack of agreement between pTRG and mrTRG could be also attributed to the differences in defining near-complete clinical response after nCRT [25]. Although mrTRG has not been proven to accurately predict pTRG, there are numerous studies indicating its value in the prediction of overall survival [26]. Finally, several studies confirmed that poor response on MRI excludes good nCRT response [11,20,27,28]. Another important issue which was pointed out by several prior studies is the fact that nCRT’s effectiveness is time-dependent. In this setting, the ongoing STAR-TREC trial suggests that restaging should be performed after 16–20 weeks [29,30]. Thus, an adequate restaging interval could lead to better agreement between mrTRG and pathology exam.

In order to improve the potential of MRI in the assessment of nCRT response, several studies evaluated the combination of different MRI parameters in the prediction of pTRG [31,32,33,34,35]. Thus, Hotker et al. investigated the combination of mrTRG and T2-weighted imaging, DWI, and DCE-MRI, respectively. They found strong correlation between T2-weighted imaging and mrTRG with pTRG in only one radiologist, but not the others involved in this assessment [31]. Moreover, Chandramohan et al. determined a high diagnostic accuracy of DWI combined with T2-weighted high-resolution MRI in the prediction of the complete response to nCRT [32]. Aiming to distinguish complete responders and non-complete responders after nCRT, Rengo et al. developed and validated a decision support model employing data-mining algorithms based on morphologic features derived from MRI images. The authors selected mrTRG, staging volume, tumor volume reduction rate, and signal intensity reduction rate for the algorithm’s development and found a sensitivity of 85.71% and a 100% specificity in accurate classification of the patients [33]. In a prospective study on 126 patients, Hall et al. concluded that addition of DWI to mrTRG improved both the sensitivity and specificity in the assessment of the complete response [34]. Similarly, Xu et al. combined mrTRG and apparent diffusion coefficient to predict a pathological CR after nCRT in patients with LARC and concluded that this combination could be a good, non-invasive method to predict the therapy response [35]. The significance of various textural parameters derived from T2-weighted imaging and ADC were pointed out in the study by Azamat et al. who showed that skewness obtained from T2-weighted imaging, decrease of T2-weighted signal intensity and ADC changes are indicators of a good response [36]. Further studies which would combine abovementioned features could possibly achieve promising results. Compared to other studies, our proposed scoring system consists of only two parameters: mrTRG and T2-weighted tumor volume regression grade. Inclusion of the T2-weighted tumor volume regression grade, the parameter whose significance in response prediction in LARC patients is well-documented, could allow overcoming of certain shortcomings of mrTRG as a single prognostic value of nCRT response. Moreover, tumor volume regression grade addition could improve the reported low inter-observer agreement of mrTRG. Diagnostic performance of newly developed prediction score in prediction of the response to nCRT showed high sensitivity, PPV, specificity, and NPV of 81.8%, 56.3%, 71.4% and 89.7%, respectively. Furthermore, 80% of patients with a high-response score (tumor volume regression ≥75% and mrTRG 1,2) in our newly proposed system were complete responders. On the other hand, 89.7% of the subjects with low-response scores were actual non-responders which indicates very good correlation of these preoperative MRI variables and pTRG. However, additional studies should be conducted to examine the learning curve of the newly proposed scoring system.

Our study has a few limitations. First, the study population consists of a relatively small number of patients. Second, the results of this study represent a single-center experience. Finally, we did not include the interval between the first MR examination and surgery, as well as post-CRT MR screening.

## 5. Conclusions

T2-weighted volumetry may increase the value of MRI in the assessment of nCRT response in patients with LARC. The results of the current study have shown that determination of tumor volume regression grade has high diagnostic accuracy for prediction of pTRG after nCRT in these patients. Moreover, a proposed novel scoring system, based on tumor volume reduction rate and mrTRG, may help clinicians in determining a more individualized approach for patients with LARC. However, future studies should validate the proposed scoring system in different populations and in everyday clinical practice.

## Figures and Tables

**Figure 1 diagnostics-13-03226-f001:**
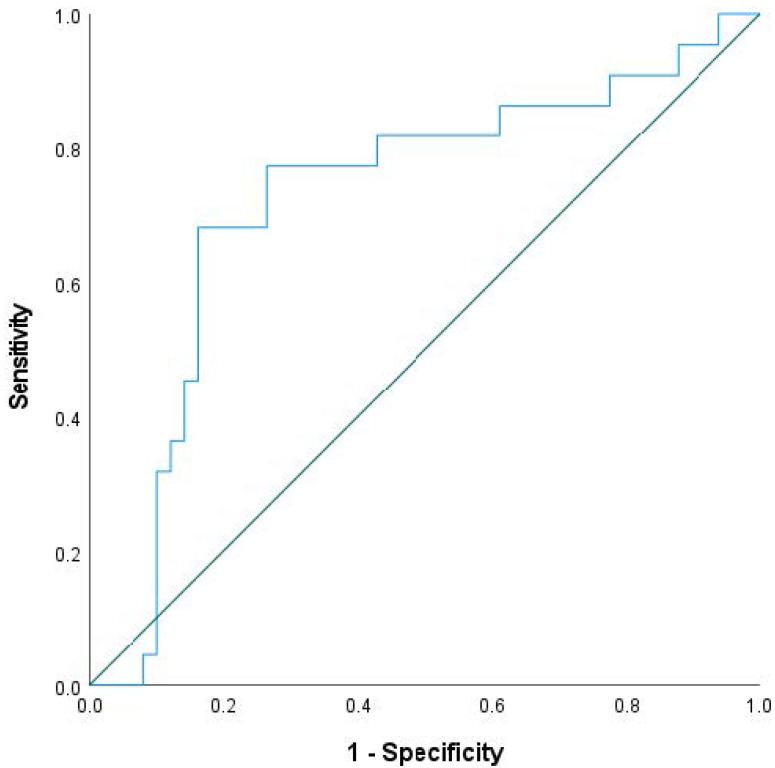
ROC curve of tumor volume regression for differentiation of responders from non-responders.

**Figure 2 diagnostics-13-03226-f002:**
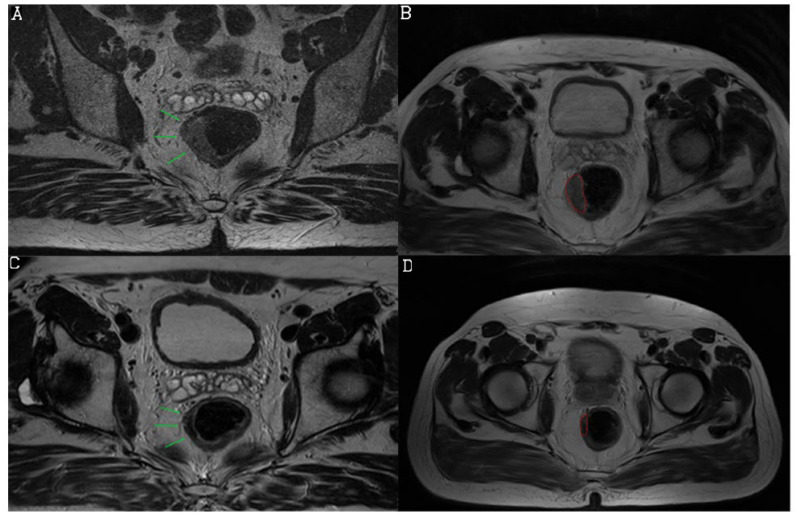
Initial and follow-up MRI examination preformed in a 63-year-old male patient. T2-weighted sequence, axial view, shows semicircular tumor in distal rectum before treatment (arrows) (**A**) and its tumor volume delineation (**B**). Follow-up MRI examination reveals reduction of tumor size with low signal intensities due to dense fibrosis (arrows) (**C**). We suggested mrTRG2 (responder, 1 point). Volume was calculated on both initial and follow-up MRI, by manually drawing a region of interest (ROI) on each tumor slice based on morphological criteria (**B**,**D**). Volumetry showed decrease from 4128.1 cm^3^ to 1632 cm^3^ which was <75% (0 point), and the prediction chance was scored as medium, with near equal chances for responders and non-responders. Histopathology revealed non-responder with pTRG3.

**Figure 3 diagnostics-13-03226-f003:**
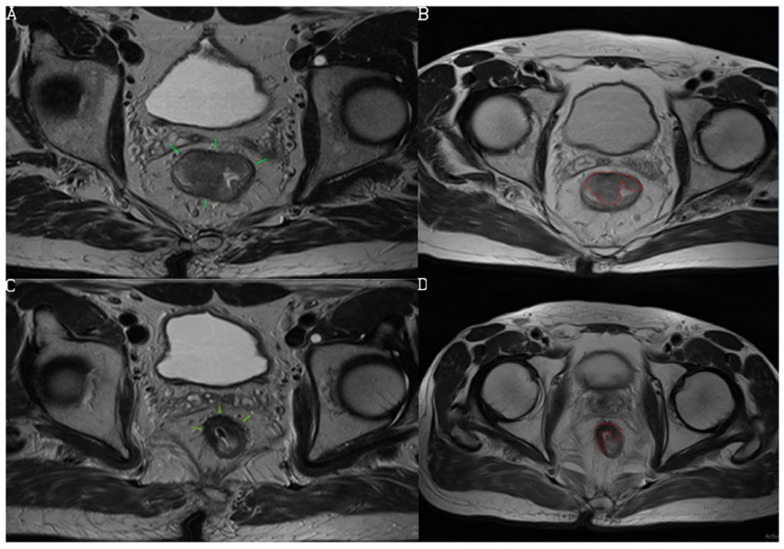
Initial and follow-up MRI examination preformed in a 59-year-old female patient. T2-weighted sequence, axial view, shows rectal mass (arrows) (**A**) and its tumor volume delineation (**B**). Follow-up MRI examination shows reduction of tumor size with low signal intensities due to dense fibrosis but with present slight zones of inhomogeneous intensities (arrows) (**C**). We suggested mrTRG2 (responder, 1 point). Volume was calculated on both initial and follow-up MRI, by manually drawing a region of interest (ROI) on each tumor slice based on morphological criteria (**B**,**D**). Volumetry showed decrease from 21,796.4 cm^3^ to 2368.7 cm^3^ which was ≥75% (1 point), and with the total score of 2, the prediction chance for responder was scored as high. Histopathology revealed complete responder with pTRG1.

**Figure 4 diagnostics-13-03226-f004:**
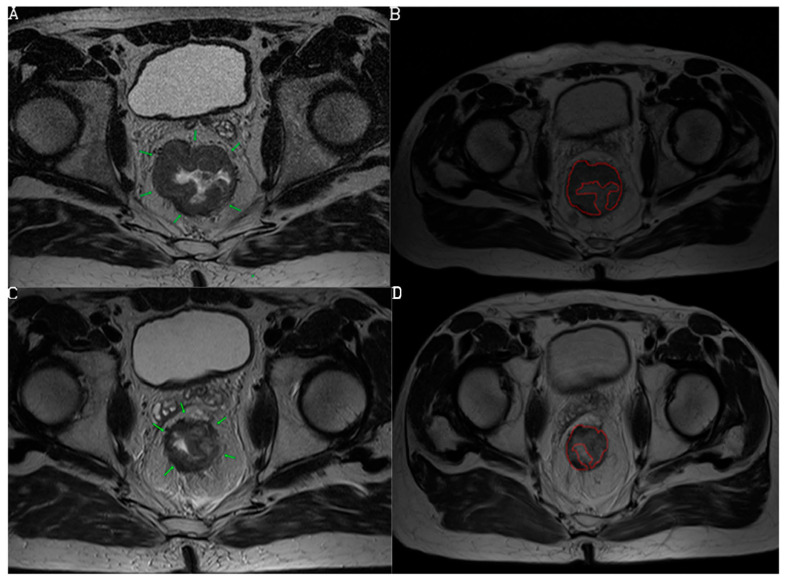
Initial and follow-up MRI examination preformed in a 71-year-old female patient. T2-weighted sequence, axial view, shows rectal mass (arrows) (**A**) and its tumor volume delineation (**B**). Follow-up MRI examination shows some reduction of tumor size with inhomogeneous signal intensities and irregular contours (arrows in (**C**)). We suggested mrTRG3 (non-responder, 0 point). Volume was calculated on both initial and follow-up MRI by manually drawing a region of interest (ROI) on each tumor slice based on morphological criteria (**B**,**D**). Volumetry showed decrease from 32,365.6 cm^3^ to 29,668 cm^3^ which was <75% (0 point), and with the total score of 0, the prediction chance for responder was scored as low. Histopathology revealed non-responder with pTRG4.

**Figure 5 diagnostics-13-03226-f005:**
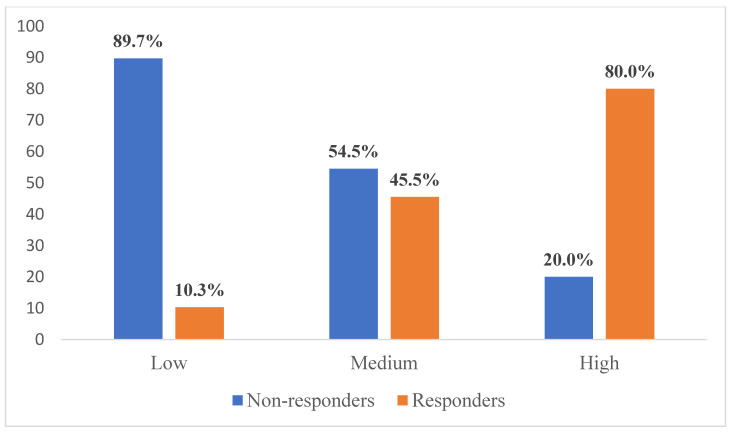
Distribution of nCRT response prediction score.

**Table 1 diagnostics-13-03226-t001:** Patient characteristics.

Variable	*n* = 71
Age, mean ± SD	61.5 ± 11.4
Gender (male), *n* (%)	47 (66.2)
Localization, *n* (%)	
Lower third	35 (49.3)
Middle third	21 (29.6)
Upper third	15 (21.1)
T stage before CRT, *n* (%)	
T1/2	3 (4.2)
T3ab	38 (53.5)
T3cd	18 (25.4)
T4a	4 (5.6)
T4b	8 (11.3)
N stage before CRT, *n* (%)	
N0	3 (4.2)
N1	28 (39.4)
N2	40 (56.3)
mrTRG, *n* (%)	
1	6 (8.5)
2	12 (16.9)
3	32 (45.1)
4	19 (26.8)
5	2 (2.8)
pTRG, *n* (%)	
1	13 (18.3)
2	9 (12.7)
3	18 (25.4)
4	26 (36.6)
5	5 (7.0)
MRF invasion before nCRT, *n* (%)	25 (35.2)
T stage after nCRT, *n* (%)	
T0	6 (8.5)
T1/2	17 (23.9)
T3ab	27 (38.0)
T3cd	13 (18.3)
T4a	3 (4.2)
T4b	5 (7.0)
N stage after nCRT, *n* (%)	
N0	48 (67.6)
N1	16 (22.5)
N2	7 (9.9)
MRF invasion after nCRT, *n* (%)	14 (19.7)

NOTE: The results are expressed as numbers (%). CRT—chemoradiotherapy; mrTRG—magnetic resonance tumor regression grade; pTRG—pathological tumor regression grade; MRF—mesorectal fascia.

**Table 2 diagnostics-13-03226-t002:** mrTRG and MRF according to the pTRG

	Total (*n* = 71)	pTRG		*p*
		Responders (*n* = 22)	Non-Responders (*n* = 49)	
mrTRG, *n* (%)	18 (25.4)	11 (50.0)	7 (14.3)	0.001
MRF invasion after nCRT, *n* (%)	14 (19.7)	1 (4.5)	13 (26.5)	0.031

mrTRG—magnetic resonance tumor regression grade; MRF—mesorectal fascia; pTRG—pathological tumor regression grade. nCRT—neoadjuvant chemioradiotherapy.

**Table 3 diagnostics-13-03226-t003:** Tumor volume regression according to the pTRG.

	Total (*n* = 71)	pTRG		*p*
		Responders (*n* = 22)	Non-Responders (*n* = 49)	
Tumor volume before nCRT, median (25th–75th percentile)	12,998.0 (9920.4–24,552.3)	14,471.3 (7897.2–23,717.2)	12,952.9 (10,470.3–25,306.7)	0.691
Tumor volume after nCRT, median (25th–75th percentile)	4670.3 (2249.0–9075.2)	2686.1 (1964.7–6502.1)	5306.2 (2924.1–11,679.0)	0.019
Tumor volume regression, median (25th–75th percentile)	66.4 (50.1–81.0)	79.9 (69.9–88.5)	63.3 (44.7–73.0)	0.003

NOTE: Data are expressed as means ± standard deviation. pTRG—pathological tumor regression grade. nCRT—neoadjuvant chemoradiotherapy.

**Table 4 diagnostics-13-03226-t004:** Diagnostic performance of tumor volume regression for differentiation of responders from non-responders.

Tumor Volume Reduction	Sensitivity (%)	PPV (%)	Specificity (%)	NPV (%)
68%	77.3	50.0	65.3	86.5
70%	77.3	51.5	67.3	86.8
72%	72.7	52.5	73.5	85.7
75%	68.2	62.5	81.6	85.1

NOTE: The results are expressed as numbers (%). PPV—positive predictive value; NPV—negative predictive value.

**Table 5 diagnostics-13-03226-t005:** nCRT response prediction score using cut-off values for tumor volume regression grade and mrTRG response to nCRT.

Tumor Volume Regression	mrTRG	
	3–5	1–2
<75%	0 (low)	1 (medium)
≥75%	1 (medium)	2 (high)

nCRT—neoadjuvant chemoradiotherapy; mrTRG—magnetic resonance tumor regression grade.

**Table 6 diagnostics-13-03226-t006:** Diagnostic performance of newly developed prediction score for differentiation of responders from non-responders.

Score	Sensitivity (%)	PPV (%)	Specificity (%)	NPV (%)
Medium/high	81.8	56.3	71.4	89.7

PPV—positive predictive value; NPV—negative predictive value.

## Data Availability

Not applicable.

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
