# Peer review of "MRI Tumor Regression Grade Combined with T2-Weighted Volumetry May Predict Histopathological Response in Locally Advanced Rectal Cancer following Neoadjuvant Chemoradiotherapy—A New Scoring System Proposal"

_diagnostics, 2023, doi:10.3390/diagnostics13203226_

Round 1

Reviewer 1 Report

The authors present a study on the value of T2-weighted volumetry as a predictor of response in locally advanced rectal cancer. The study is well designed, the statistical analysis is correctly chosen and applied. Literature is up to date.

However I have some minor remarks.

I advise to change the word „pathological” into „histopathological” in the title. 

It is a limitation that evaluation was made in consensus by the readers. The other is a small number of patients (also acknowledged by the authors).  Would you care to explain what does retrospective-prospective design of the study mean?

The time period of the study should be revised. In materials and methods section, the authors state „period from January 2020. to December 2023”.   It is not yet the end of July 2023. What was the reason for using an external application for volumetry, if I understood correctly? In order for it to be incorporated into clinical practice it would be better to have it available within the software in use everyday. 

The authors present a study on the value of T2-weighted volumetry as a predictor of response in locally advanced rectal cancer. The study is well designed, the statistical analysis is correctly chosen and applied. Literature is up to date. 

However I have some minor remarks.

I advise to change the word „pathological” into „histopathological” in the title. 

It is a limitation that evaluation was made in consensus by the readers. The other is a small number of patients (also acknowledged by the authors).  Would you care to explain what does retrospective-prospective design of the study mean?

The time period of the study should be revised. In materials and methods section, the authors state „period from January 2020. to December 2023”.   It is not yet the end of July 2023. What was the reason for using an external application for volumetry, if I understood correctly? In order for it to be incorporated into clinical practice it would be better to have it available within the software in use everyday. 

Author Response

Response to Reviewer 1 Comments

Thank you very much for your comments and suggestions on how to improve the quality of this manuscript. Please find the detailed responses below and the corresponding revisions/corrections highlighted in track changes in the re-submitted file.

The authors present a study on the value of T2-weighted volumetry as a predictor of response in locally advanced rectal cancer. The study is well designed, the statistical analysis is correctly chosen and applied. Literature is up to date.

However I have some minor remarks.

Comment 1: I advise to change the word „pathological” into „histopathological” in the title. 

Response 1: Thank you for the suggestion. We have made the requested change.

Comment 2: It is a limitation that evaluation was made in consensus by the readers. The other is a small number of patients (also acknowledged by the authors). 

Response 2: In order to improve diagnostic accuracy of radiological examinations it is a standard procedure to perform evaluation in consensus of two readers. Furthermore, in majority of recent radiological studies it is advisable to have consensus of two readers. Concerning the number of patients, due to the pandemic, the number of patients referred to our clinic was reduced and not all of them could receive all the necessary examinations in adequate time intervals.

Comment 3: Would you care to explain what does retrospective-prospective design of the study mean?

Response 3: Given that the research started in April 2021, a part of the MRI studies were conducted prior to this period, specifically from January 2020. These examinations are classified as "retrospective", as data were extracted after the MRI studies. Conversely, the remaining MRI studies were conducted after April 2021, extending until the conclusion of 2022. These MRI studies are categorized as "prospective", as data were extracted at the time of MRI studies. To enhance clarity for the readers, the study design has been now modified to a cohort study. All consecutive patients were enrolled in the study, undergoing a routine MRI examination with consistent regimens for neoadjuvant chemoradiotherapy.

Comment 4: The time period of the study should be revised. In materials and methods section, the authors state „period from January 2020. to December 2023”. It is not yet the end of July 2023.

Response 4: We thank the reviewer for this remark. We have now corrected this into "December 2022" which represents the correct information.

Comment 5: What was the reason for using an external application for volumetry, if I understood correctly? In order for it to be incorporated into clinical practice it would be better to have it available within the software in use everyday. 

Response 5: The decision was made to use an external tool for volumetry due to its notable features, including user-friendly interface, high precision, and widespread availability as a free downloadable resource.

Reviewer 2 Report

In the abstract: Please provide a clearer background on the utility of MRI in predicting pathological response. Also, ensure the conclusion is in the same font size.

In the introduction, please elaborate on the T2-weighted volumetry measure, why it's better than other MRI techniques for measuring response to neoadjuvant treatment, and explain the gold standard MRI for rectal cancer.

Methods: Revise the document to use "histopathological analysis" instead of "pathohistological." Clarify which part of the analysis is retrospective and which is prospective, and explain why MRI analysis is done two weeks after neoadjuvant treatment. The inclusion and exclusion criteria should be clearer, and control MRI should be defined. Number 5 is repeated.

Clarify the meaning of inadequate quality of any mentioned MR examinations. If surgery is included in the inclusion criteria, explain why MRI is done 6-8 weeks after chemo-radiotherapy, and provide a diagram or explanation of the MRI timing, and give the rationale. 

Results: tables are confused because the research question is confusing 

Abstract: Please be more apparent in the background about MRI's utility in predicting pathological response. Conclusions are in another size of the letter.

Introduction: Please describe in more detail what implies the T2-weighted volumetry measure, why it could be a better way to measure the response to neoadjuvant treatment over other MRI techniques and describe in more detail which is the gold standard of MRI for rectal cancer.

Methods: Please revise the edition of the document. For example, in number 5, you have THE. It is better to use histopathological analysis versus pathohistological. Please describe in more detail which part of the analysis is retrospective and which part is prospective; why did you decide two weeks after the neoadjuvant treatment for the MRI analysis? Inclusion criteria and exclusion criteria need to be clarified. What mean control MRI in patients without cancer? 5 number is repeated, which means the quality of any mentioned MR examinations needs to be improved. If the inclusion criteria include patients with surgery, why did you mention that you do the MRI six to eight weeks after chemo-radiotherapy? Please give a diagram or clarify the times or MRI.

In most methods, evaluating the possibility of anus preservation for basal and pre-surgery measures is important. Post-operative MRI should include the circumferential resection margin analysis and MR imaging for vascular invasions, such as EMVI (extramural vascular invasion) on T2-weighted images. Results should include information on mesolectal fascia (MRF) and pathological response, which are part of the analysis.

 Discussion must also cover these parameters.

English must be improved. 

Author Response

Response to Reviewer 2 Comments

Thank you very much for your comments and suggestions on how to improve the quality of this manuscript. Please find the detailed responses below and the corresponding revisions/corrections highlighted in track changes in the re-submitted file.

Comment 1: In the abstract: Please provide a clearer background on the utility of MRI in predicting pathological response. Also, ensure the conclusion is in the same font size.

Response 1: Thank you for the suggestion. We have now modified Abstract.

Comment 2: In the introduction, please elaborate on the T2-weighted volumetry measure, why it's better than other MRI techniques for measuring response to neoadjuvant treatment, and explain the gold standard MRI for rectal cancer.

Response 2: MRI is imaging procedure which provides excellent insight into anatomy of rectal wall, mesorectal fat and other pelvic structures. Therefore, MRI is considered to be the gold standard for the initial examination of patients with histopathologically proven rectal cancer. Unfortunately, it is well known that accuracy of MRI dramatically decreases after nCRT as it is often hard or even impossible to make a difference between signal intensities of the therapy-induced fibrosis and viable tumor cells. That is the reason why other MRI parametars than those included in the standard protocol are needed in everyday clinical practice for assessment of nCRT response in patients with rectal cancer. In that setting, our study suggests that T2-weighted volumetry combined with mrTRG based on standard MRI protocol may improve the accuracy of MRI. The advantages and disadvantages of MRI in rectal cancer diagnosis are elaborated in details in the manuscript (Pg:2, Ln:14). Moreover, the data from the recent literature considering the value of T2-weighted volumetry are already mentioned in the manuscript (Pg:2, Ln:29)

Comment 3: Please revise the edition of the document. For example, in number 5, you have THE.

Response 3: Corrected in revised manuscript. Thank you for the suggestion.

Comment 4: Methods: Revise the document to use "histopathological analysis" instead of "pathohistological."

Response 4: Thank you for the comment. Corrected in revised manuscript.

Comment 5: Please describe in more detail which part of the analysis is retrospective and which part is prospective.

Response 5:

Given that the research started in April 2021, a part of the MRI studies were conducted prior to this period, specifically from January 2020. These examinations are classified as "retrospective", as data were extracted after the MRI studies. Conversely, the remaining MRI studies were conducted after April 2021, extending until the conclusion of 2022. These MRI studies are categorized as "prospective", as data were extracted at the time of MRI studies. To enhance clarity for the readers, the study design has been now modified to a cohort study. All consecutive patients were enrolled in the study, undergoing a routine MRI examination with consistent regimens for neoadjuvant chemoradiotherapy.

Comment 6: Why did you decide two weeks after the neoadjuvant treatment for the MRI analysis?

Response 6: Two weeks’ time frame is recommended after initial biopsy and histopathology confirmation, not the neoadjuvant treatment. The reason is to avoid over staging as a result of bleeding or other early post-biopsy sequelae.

Comment 7: Inclusion criteria and exclusion criteria need to be clarified.

Response 7: Inclusion criteria are defined and explained on the basis of previous research and include: 1) pathohistological confirmation of rectal cancer; 2) patients treated with neoadjuvant chemoradiotherapy; 3) both initial and control MRI examination performed; and 4) patients who underwent surgery after neoadjuvant chemoradiotherapy therapy. The exclusion criteria were defined in correlation with previously established protocols. The exclusion criterion “technically inadequate quality of any of the mentioned MR examinations” was explained in the response 8.

Comment 8: Clarify the meaning of inadequate quality of any mentioned MR examinations. If surgery is included in the inclusion criteria, explain why MRI is done 6-8 weeks after chemo-radiotherapy, and provide a diagram or explanation of the MRI timing, and give the rationale. 

Response 8: Usually, inadequate quality of MRI examination is the result of artefacts (e.g. motion artefacts) and interpretation of those MRI sequencies is not accurate. Generally, it is advised to perform surgery 6-8 weeks after nCRT, and delaying even more than 8 weeks may be benefitial. This could be explained by the fact that many previous studies (Seo N, Kim H, Cho MS, Lim JS. Response Assessment with MRI after Chemoradiotherapy in Rectal Cancer: Current Evidences. Korean J Radiol. 2019;20(7):1003-1018.; Goodman KA. Timing Is Everything: What Is the Optimal Duration After Chemoradiation for Surgery for Rectal Cancer? J Clin Oncol. 2016;34(31):3724-3728.) have shown that this time is necessary for the radiation therapy to exert an effect. Moreover, some side effects of radiation therapy, such as edema, could impair the appropriate interpretation of nCRT response. Therefore, in accordance with widely accepted protocols, post-nCRT MRI in our study was  performed 6-8 weeks after the treatment and right before surgery.

Comment 9: Results: tables are confused because the research question is confusing 

Response 9: We have revised Results section in both Abstract and Manuscript. Hopefully, this will clarify our research findings.

Comment 10: In most methods, evaluating the possibility of anus preservation for basal and pre-surgery measures is important. Post-operative MRI should include the circumferential resection margin analysis and MR imaging for vascular invasions, such as EMVI (extramural vascular invasion) on T2-weighted images.

Response 10: Regarding circumferential resection margine, we did not perform post-operative MRI for this study. EMVI and MRF involvement are as you mentioned very important features which should be routinely evaluated on both initial and posttreatment MRI examination. The EMVI analysis, as explained in Material and Methods section, is a part of a our wider study with numerous other parameters. However, as in our current research we focused on combining mrTRG and MR volumetry, and proposal of new scoring system, it is beyond the topic and the scope of this manuscript to include all relevant data about EMVI. Nevertheless, we hope that these results and detailed EMVI and MRF analysis could be a part of our next study.

Comment 11: Results should include information on mesolectal fascia (MRF) and pathological response, which are part of the analysis.

Response 11: We have now included this information in new Table 2.

Comment 12:  Discussion must also cover these parameters.

Response 12: The same response as in comment 10.

Reviewer 3 Report

Dear Author(s),

Please edit your manuscript to include the following changes:

- The article's title needs to be revised to be more succinct and consistent with the goal that guided the design of the current study.

- In the results section of the abstract of the present study, make an effort to highlight the most significant findings without going into excessive detail.

- I advise the author(s) to include a paragraph in the section on statistical analysis that explains why the statistical approach presented here was utilized to analyze the study's findings.

- Rewriting the study's conclusion in light of whether the current study met its goals, that is, whether the research problem was resolved.

- The existing study's references need to be updated with more recent ones because some of them are old. I recommended that the author(s) restrict their references to the years 2023 and the five years prior.

Good luck,

Author Response

Response to Reviewer 3 Comments

Thank you very much for your comments and suggestions on how to improve the quality of this manuscript. Please find the detailed responses below and the corresponding revisions/corrections highlighted in track changes in the re-submitted file.

Dear Author(s), please edit your manuscript to include the following changes:

Comment 1: The article's title needs to be revised to be more succinct and consistent with the goal that guided the design of the current study.

Response 1: Thank you for the suggestion. We changed the title to: The combination of MRI tumor regression grade and T2-weighted tumor volumetry may add in prediction of histopathological response in locally advanced rectal cancer after neoadjuvant chemoradiotherapy – A new scoring system proposal

Comment 2: In the results section of the abstract of the present study, make an effort to highlight the most significant findings without going into excessive detail.

Response 2: Thank you for the suggestion. We have now modified Abstract to highlight the most significant findings.

Comment 3: I advise the author(s) to include a paragraph in the section on statistical analysis that explains why the statistical approach presented here was utilized to analyze the study's findings.

Response 3: Thank you for the suggestion. We have now explained statistical analysis in more details.

Comment 4: Rewriting the study's conclusion in light of whether the current study met its goals, that is, whether the research problem was resolved.

Response 4: Corrected in revised manuscript.

Comment 5: The existing study's references need to be updated with more recent ones because some of them are old. I recommended that the author(s) restrict their references to the years 2023 and the five years prior.

Response 5: Thank you for your comment. We gave our best to include majority of relevant publications that are up to date, from the past 5 years. However, regarding specific topic, we could not avoid including some of the relevant publications from older dates, too.

Round 2

Author Response

REVIEWER 2

Comment 1: Despite the authors made an effort to improve the title, the title remains excessively lengthy and unappealing. It must be enhanced for better impact and effectiveness.

Answer 1: We have further modified the title according to the suggestions of the reviewer: “MRI tumor regression grade combined with T2-weighted volumetry may predict histopathological response in locally advanced rectal cancer following neoadjuvant chemoradiotherapy – A new scoring system proposal”

Comment 2: “MRI is imaging procedure which provides excellent insight into anatomy of rectal wall, mesorectal fat and other pelvic structures. Therefore, MRI is considered to be the gold standard for the initial examination of patients with histopathologically proven rectal cancer. Unfortunately, it is well known that accuracy of MRI dramatically decreases after nCRT as it is often hard or even impossible to make a difference between signal intensities of the therapy-induced fibrosis and viable tumor cells. That is the reason why other MRI parametars than those included in the standard protocol are needed in everyday clinical practice for assessment of nCRT response in patients with rectal cancer. In that setting, our study suggests that T2-weighted volumetry combined with mrTRG based on standard MRI protocol may improve the accuracy of MRI.”

This paragraph or idea must be on your document, is the main idea of your work, and was well explained in your answer, but unfortunately, it is not clear in your paper.

Answer 2: We thank the reviewer for this remark and apologize for this inconvenience. We have now revised this paragraph in our manuscript according to the suggestions.

Comment 3: The advantages and disadvantages of MRI in rectal cancer diagnosis are elaborated in details in the manuscript (Pg:2, Ln:14). Moreover, the data from the recent literature considering the value of T2-weighted volumetry are already mentioned in the manuscript (Pg:2, Ln:29)

The advantages and drawbacks of T2-weighted volumetry and mrTRG are not fully understood.

For example, in their paper, Azamat notes the limitations of mrTRG and proposes a new evaluation method that involves textural features obtained from T2 weighted imaging and ADC maps.

Answer 3: We want to thank the reviewer for this important remark.

Azamat et al used textural parameters derived from T2-weighted imaging and ADC. The authors analyzed those features separately and showed that skewness obtained from T2-weighted imaging, decrease of T2-weighted signal intensity and ADC changes are significant indicators of a good response. Texture analysis as an integral part of Radiomics is the cornerstone of many recent researches but it still requires quite complex software analysis.
Our scoring sistem represents a non-invasive, low-cost, and easily feasible
measure in everyday clinical practice with excellent performance, since 80% of patients
with a high-response score were complete responders, and 89.7% of the subjects with low-response scores were actual non-responders.
We have added a paragraph about the advantages and drawbacks of T2-weighted volumetry and ADC, using the informations from kindly shared paper by Azamat et al.

Comment 4: Please give us more detail that why your proposal is superior and its benefits. More information can be found at https://www.ncbi.nlm.nih.gov/pmc/articles/PMC9720879/.

Answer 4: Compared to other studies, our proposed scoring system consists of only two parameters: mrTRG and T2-weighted tumor volume regression grade. Inclusion of the T2-weighted tumor volume regression grade, the parameter whose significance in response prediction in LARC patients is well-documented, could allow us to overcome certain shortcomings of mrTRG as a single prognostic value of nCRT response. Moreover, tumor volume regression grade addition could overcome the reported low inter-observer agreement of mrTRG.

Additionally, this scoring system represents a non-invasive, low-cost, and easily feasible

measure in everyday clinical practice with excellent performance, since 80% of patients

with a high-response score were complete responders, and 89.7% of the subjects with low-response scores were actual non-responders.

Finally, we believe that a scoring system with the mentioned two parameters would enable easier reproducibility and use by younger radiologists. However, additional studies should be conducted to examine the learning curve of the newly proposed scoring system. The corrections and detailed explanation was added in the revised manuscript according to the suggestions of the reviewer.

Comment 5: Please define Scoring systems established by the Magnetic Resonance Imaging and Rectal Cancer European Equivalence Study (MERCURY) study group and the European Society of Gastrointestinal and Abdominal Radiology (ESGAR), and where are their limitations, and why your proposal is better. See paper

https://www.ncbi.nlm.nih.gov/pmc/articles/PMC6609432/ that explains in detail.

Comment 5: We thank the reviewer for this important remark. We apologize for not including these details in our methodology. In our study, the mrTRG was classified according to the MERCURY study group and we have added a detailed explanation in the Methodology section.

„The MRI examinations were reviewed in consensus by two radiologists with 8 and 13 years of expertise in rectal cancer MRI. The following parameters were analyzed: tumor localization, craniocaudal tumor diameter, presence of extramural tumor propagation, lympho-nodal status, assessment of tumor infiltration of the mesorectal fascia (MRF), assessment of extramural vascular invasion according to Smith, local tumor stage before and after nCRT, and MRI estimated tumor regression grade (mrTRG). We used a scoring system established by the Magnetic Resonance Imaging and Rectal Cancer European Equivalence Study (MERCURY) study group for mrTRG (Battersby NJ, How P, Moran B, Stelzner S, West NP, Branagan G, Strassburg J, Quirke P, Tekkis P, Pedersen BG, Gudgeon M, Heald B, Brown G; MERCURY II Study Group. Prospective Validation of a Low Rectal Cancer Magnetic Resonance Imaging Staging System and Development of a Local Recurrence Risk Stratification Model: The MERCURY II Study. Ann Surg. 2016;263(4):751-60.). The construction of mrTRG is based on pathologic TRG (pTRG). The classification of mrTRG relies on the relative prevalence of fibrous or tumor signal intensity within the tumor.In the context of radiology, Grade 1 signifies a comprehensive radiologic response characterized by the presence of a linear or crescentic scar, as observed through MRI. Grade 2 denotes a favorable response, wherein MRI findings reveal dense fibrosis without any discernible residual tumor, thereby suggesting either minimal residual disease or the absence of a tumor. Grade 3 corresponds to a moderate response, wherein more than 50% of the areas exhibit fibrosis or mucin, accompanied by visible intermediate tumor signal on MRI. Grade 4 indicates a marginal response to treatment, with MRI findings indicating the presence of a few areas with fibrosis or mucin, predominantly displaying tumor-derived MRI signals. Lastly, Grade 5 signifies a lack of response to therapy, characterized by a tumor similar to the baseline or significant regrowth of the tumor. “

Besides the MERCURY study group, the European Society of Gastrointestinal and Abdominal Radiology (ESGAR) consensus meetings have proposed a classification system consisting of three stages: a wall that is entirely normalized, only fibrotic wall thickening, and a residual mass (Beets-Tan RGH, Lambregts DMJ, Maas M, Bipat S, Barbaro B, Curvo-Semedo L, Fenlon HM, Gollub MJ, Gourtsoyianni S, Halligan S, Hoeffel C, Kim SH, Laghi A, Maier A, Rafaelsen SR, Stoker J, Taylor SA, Torkzad MR, Blomqvist L. Magnetic resonance imaging for clinical management of rectal cancer: Updated recommendations from the 2016 European Society of Gastrointestinal and Abdominal Radiology (ESGAR) consensus meeting. Eur Radiol. 2018;28(4):1465-1475.). The authors put up the suggestion that the presence of a normalized two-layered wall following chemoradiotherapy (CRT) may indicate complete response (CR) on T2-weighted imaging. Additionally, they argue that the absence of an isointense mass, accompanied by fibrotic residue, could indicate either CR or near-CR. Given the rare incidence of complete normalization of the rectal wall following chemoradiotherapy (CRT) in clinical settings, the ESGAR group appears to adopt a more rigorous approach in characterizing radiologic complete response (CR). Therefore, we used the MERCURY classification system to define mrTRG in our study.

Comment 6: “We proposed a new scoring system that could aid in distinguishing responders to nCRT from non-responders, as the first mentioned group of these patients could benefit from organ-preserving treatment and “watch and wait” strategy.”

Then you have this in your introduction as the aim.

“Therefore, the aim of this study was to investigate the potential benefits of combining MRI-based T2-weighted volumetry and mrTRG in prediction of pathological response in patients with LARC after receiving nCRT and to propose the new MRI-based scoring system.”

The new, or a new proposal, what includes this proposal? Why is better than others?

Please detail, which are the elements of this scoring system, and the methodology, please detail the steps that have to repeat your proposal. On the discussion give us the advantage of your proposal over…. Others.

Answer 6: Thank you for your remarks. The answer to this comment is already given in the Answer No4 since the comment regarding benefits of proposed scoring system is very similar.

Comment 7: Methodology: the authors answered:

“Given that the research started in April 2021, a part of the MRI studies were conducted prior to this period, specifically from January 2020. These examinations are classified as "retrospective", as data were extracted after the MRI studies. Conversely, the remaining MRI studies were conducted after April 2021, extending until the conclusion of 2022. These MRI studies are categorized as "prospective", as data were extracted at the time of MRI studies. To enhance clarity for the readers, the study design has been now modified to a cohort study. All consecutive patients were enrolled in the study, undergoing a routine MRI examination with consistent regimens for neoadjuvant chemoradiotherapy.”

But they do not change on the methods: they have the same information that have before.  

“This was a retrospective-prospective study which included seventy-one patients with pathohistopathologically confirmed rectal cancer in a period from January 2020. to December 20223. The study was approved by the relevant institutional review board, and written informed consent was obtained from all patients. “

Answer 7: We apologize to the reviewer for inconvenience. We have now corrected this in manuscript.

Comment 8: This is an answer not detailed on the methodology and improve the work.

Usually, inadequate quality of MRI examination is the result of artefacts (e.g. motion artefacts) and interpretation of those MRI sequencies is not accurate. Generally, it is advised to perform surgery 6-8 weeks after nCRT, and delaying even more than 8 weeks may be benefitial. This could be explained by the fact that many previous studies (Seo N, Kim H, Cho MS, Lim JS. Response Assessment with MRI after Chemoradiotherapy in Rectal Cancer: Current Evidences. Korean J Radiol. 2019;20(7):1003-1018.; Goodman KA. Timing Is Everything: What Is the Optimal Duration After Chemoradiation for Surgery for Rectal Cancer? J Clin Oncol. 2016;34(31):3724-3728.) have shown that this time is necessary for the radiation therapy to exert an effect. Moreover, some side effects of radiation therapy, such as edema, could impair the appropriate interpretation of nCRT response. Therefore, in accordance with widely accepted protocols, post-nCRT MRI in our study was performed 6-8 weeks after the treatment and right before surgery.

Answer 8: Thank you for the suggestions, we have now added detailed explanation in the Methodology section.

Comment 9: What is the efficacy of your proposal on nodes and the circumferential margin of your score??

Answer 9: Thank you for your question. As mentioned before, in our current research we focused on combining mrTRG and MR volumetry for prediction of response after nCRT. We did not evaluate the correlation of preoperative MRI imaging features and circumferential resection margin. The assesment of lymph nodes requires completely different approach and would be beyond the scope of the current manuscript. We hope that our results based on nodes and CRM analysis could be a part of further studies.

Comment 10: How do you get the low, medium, and high prediction change? Do you include the sum of mrTGR and tumor volume? If I try to apply to the clinic, do I have to sum both? For example, 0 of tumor volume regression under 75%, and mrTRG 1,2 is two and means high?  This is the main point of your work. Please describe in detail how to use your score. To be clearer to the readers, how do you get the score? is valid only for tumors? or also for lymph nodes?

Answer 10: Thank you for your suggestions. Table 5 is reformatted in order to be clearer for the readers. Corresponding text is also changed according to suggestions.  The prediction score is valid only for tumors.

Comment 11: Discussion: Some groups make similar efforts to yours. Please give us more rationale for why your proposal is better and why you are unique.

Answer 11: The detailed explanation is added in revised manuscript.

Compared to other studies, our proposed scoring system consists of only two parameters: mrTRG and T2-weighted tumor volume regression grade. Inclusion of the T2-weighted tumor volume regression grade, the parameter whose significance in response prediction in LARC patients is well-documented, could allow overcoming of certain shortcomings of mrTRG as a single prognostic value of nCRT response. Moreover, tumor volume regression grade addition could overcome the reported low inter-observer agreement of mrTRG.

Additionally, this scoring system represents a non-invasive, low-cost, and easily feasible measure in everyday clinical practice with excellent performance, since 80% of patients with a high-response score were complete responders, and 89.7% of the subjects with low-response scores were actual non-responders.

Finally, we believe that a scoring system with the mentioned two parameters would enable easier reproducibility and use by younger radiologists. However, additional studies should be conducted to examine the learning curve of the newly proposed scoring system.

Comment 12: Your roc curve has a low is less than 8, but is similar to others for this measure. How much does the sum of mrTRG and T2-volumetry improve this value? Do you have a roc curve?

Answer 12: Sum of mrTRG and T2-volumetry improves ROC Area Under the Curve (AUC) to 0.801 and this info is added in the manuscript text. ROC curve can be added if needed.

Comment 13: On tables please leave the abbreviations meaning in the caption.

Answer 13: We have now added below the tables the caption abbreviations meaning.

Comment 14: The document contains grammatical errors that require attention.

Answer 14: We have now improved the manuscript and corrected grammatical errors.

Round 3

Reviewer 2 Report

Thanks for improving your paper. You have a grammatical mistake, as shown in Figure 6, where it is scor; you miss an e. Please review the references; reference 15 is not listed on the reference list. 

There is a grammatical mistake that needs to be addressed. 

Author Response

We thank the reviewer for his remarks and apologize for inconvenience. We have now revised our manuscript and made demanded change regarding grammatical mistake in Table 6 (missing letter in the word SCORE). References are checked and should be well listed.